# Thalassemias: An Overview

**DOI:** 10.3390/ijns5010016

**Published:** 2019-03-20

**Authors:** Michael Angastiniotis, Stephan Lobitz

**Affiliations:** 1Thalassemia International Federation, Strovolos 2083, Nicosia, Cyprus; 2Department of Pediatric Oncology/Hematology, Kinderkrankenhaus Amsterdamer Straße, 50735 Cologne, Germany

**Keywords:** thalassemia, burden of disease, newborn screening, hemoglobinopathies

## Abstract

Thalassemia syndromes are among the most serious and common genetic conditions. They are indigenous in a wide but specific geographical area. However, through migration they are spreading across regions not previously affected. Thalassemias are caused by mutations in the α (*HBA1/HBA2*) and β globin (*HBB*) genes and are usually inherited in an autosomal recessive manner. The corresponding proteins form the adult hemoglobin molecule (HbA) which is a heterotetramer of two α and two β globin chains. Thalassemia-causing mutations lead to an imbalanced globin chain production and consecutively to impaired erythropoiesis. The severity of the disease is largely determined by the degree of chain imbalance. In the worst case, survival is dependent on regular blood transfusions, which in turn cause transfusional iron overload and secondary multi-organ damage due to iron toxicity. A vigorous monitoring and treatment regime is required, even for the milder syndromes. Thalassemias are a major public health issue in many populations which many health authorities fail to address. Even though comprehensive care has resulted in long-term survival and good quality of life, poor access to essential components of management results in complications which increase the cost of treatment and lead to poor outcomes. These requirements are not recognized by measures such as the Global Burden of Disease project, which ranks thalassemia very low in terms of disability-adjusted life years (DALYs), and fails to consider that it ranks highly in the one to four-year-old age group, making it an important contributor to under-5 mortality. Thalassemia does not fulfil the criteria to be accepted as a target disease for neonatal screening. Nevertheless, depending on the screening methodology, severe cases of thalassemia will be detected in most neonatal screening programs for sickle cell disease. This is very valuable because: (1) it helps to prepare the affected families for having a sick child and (2) it is an important measure of secondary prevention.

## 1. Introduction

The hereditary disorders of the hemoglobin molecule are among the commonest of clinically serious genetic conditions [1]. They are of two general types: those in which a mutation interferes with the amount of protein produced (thalassemias), and those that result in a structural change of the hemoglobin molecule, leading to the production of a variant protein (hemoglobinopathies).

In this article we will review the pathophysiology and the clinical and public health consequences of thalassemias. These include two categories, the α- and β-thalassemias, according to which the globin chain of the hemoglobin molecule is inadequately produced. The clinically most serious conditions are the β-thalassemias in the homozygous state, while the α-thalassemia homozygotes are usually lethal in utero.

The numbers of affected patients are not known. Very few countries maintain a patient registry and in many others, children die from the more severe transfusion-dependent syndromes before they are even diagnosed. Rough estimates of expected global annual births are around 60,000 [1]. The distribution of the thalassemia genes stretches from the Mediterranean basin and Sub-Saharan Africa through the Middle East to the Far East including South China and the Pacific Islands. In northern regions, these genes are rare in the indigenous populations, but population movements, both for economic reasons and due to political instability, are contributing to a changing epidemiology [2,3]. The necessity for lifelong treatment, the prevention of serious complications through regular monitoring, and premature deaths in many patients make these disorders a significant health burden requiring public health planning and policy making [4]. This is a process which countries with few resources are often unable to follow. Even in the well-resourced countries of the West, the rarity of the condition does not always allow for expertise to develop, and optimum care is also lacking here.

## 2. Pathophysiology of Thalassemias

In the physiological state, the hemoglobin molecule is a heterotetramer consisting of two α and two non-α globin chains, each carrying a heme molecule with a central iron. In this state, the oxygen-carrying capacity of the molecule is maximal. The non-α globin chains can be β chains which coupled with α chains form adult hemoglobin (HbA), while α chains and δ chains form a minor fraction of adult hemoglobin (HbA_2_). Finally, α and γ chains form the fetal hemoglobin (HbF). The production of the globin chains is regulated by the α globin cluster on chromosome 16 with the two α globin genes *HBA1* and *HBA2*, and the β globin cluster on chromosome 11 with the genes for the γ, δ, and β globin chains. The physiological situation is characterized by a balanced production of the α and the non-α globin chains that ensures a reciprocal pairing into the normal tetramers. In the thalassemias, this equilibrium is disrupted by the defective production of one of the globin chains. Any reduced production of one of the globin chains within the developing red cell will cause an accumulation of the normally produced chain that can no longer find the equivalent amount of its heterologous partner to assemble to the normal heterotetramer. If α globin chains are not produced in adequate amounts there will be an accumulation of β globin chains (α-thalassemia); if β globin chains are inadequately produced then α globin chains will accumulate (β-thalassemia). These observations were made possible by the introduction of methods to separate and quantify these globin chains [5,6]. These studies enabled the understanding of the pathophysiology of these conditions as being the result of the chain imbalance [7].

The excess unpaired and insoluble α globin chains in β-thalassemia cause apoptosis of red cell precursors, resulting in ineffective erythropoiesis. The excess non-α globin chains in α-thalassemia assemble as γ_4_ tetramers (Hb Bart’s) in intrauterine life and β_4_ tetramers (HbH) after birth. Both of these abnormal homotetramers are poor carriers of oxygen (too high affinity for oxygen). The excess chains have further devastating effects on the function of erythrocytes and their ability to deliver oxygen [8,9].
The production of hemoglobin starts in the proerythroblast and increases during erythroid maturation through the basophilic, polychromatophilic and orthochromatic phases of red cell maturation. In erythroblasts, the excess α globin chains in β-thalassemia precipitate at the cell membrane and cause oxidative membrane damage and premature cell death by apoptosis. This happens within the erythropoietic tissue and so results in ineffective erythropoiesis [10].Some of the immature red cells pass into the circulation. Because of their membrane defect, they are fragile and prone to hemolysis. They also exhibit an altered deformability and are trapped by the spleen where they are destroyed by macrophages. This leads to an enlargement of the spleen which can become massive, leading to the development of functional hypersplenism with removal of platelets and white cells as well as red cells.Ineffective erythropoiesis, removal of abnormal cells by the spleen, and hemolysis all contribute to an anemia of variable severity.

The response to anemia is twofold:The kidneys increase secretion of erythropoietin (EPO). EPO is a cytokine that targets red cell precursors in response to the oxygen requirement of tissues. EPO secretion results in an increased red cell production, but because of the defect of erythroblast maturation this will make the ineffective erythropoiesis worse. This is a vicious cycle that results in expansion of hematopoietic tissue within the bone marrow and the destruction of bone architecture, thus contributing to bone disease and fragility. In some patients, extramedullary hematopoietic masses develop within the liver, the spleen, and the reticuloendothelial system.Hepcidin is a regulator of iron absorption [11] and produced by liver cells. It regulates the expression of ferroportin, a protein which directly facilitates enterocytic iron absorption in the gut. Independently of the cause, in severe anemia, hepcidin production is suppressed which results in increased iron absorption [12]. This contributes to iron overload, especially in patients who are not regularly transfused.

The degree of anemia is variable and depends on the mutation or combination of mutations in each individual patient. There are about 200 known mutations on the β gene cluster. Some mutations do not allow any β globin chain production. These are known as β^0^ mutations while other mutations allow some β globin chain production and are referred to as β^+^ and β^++^ mutations, respectively [13]. Likewise, in α-thalassemia more than 100 varieties have been described [14]. The degree of anemia and the severity of the clinical effect can be modified by other mitigating factors. The most common of these is the co-inheritance of factors that reduce globin chain imbalance such as when α-thalassemia is co-inherited in β-thalassemia homozygotes, resulting in a milder β-thalassemia syndrome.

The treatment of severe anemia is blood transfusion. In the serious transfusion dependent forms, regular transfusions from early childhood lead to severe iron overload. In the physiological state 1–2 mg of iron are absorbed from food sources daily and the same amount is excreted fecally. Increased gastrointestinal absorption of iron in thalassemia aggravates transfusional iron burden and results in the excess iron being taken up by proteins produced in the liver, including transferrin and ferritin. Protein bound iron is stored mainly in the liver and is not toxic. However, since each unit of transfused blood contains 100–200 mg of iron (0.47 mg/mL), in regularly transfused patients, the capacity of these proteins to bind iron is saturated soon and non-transferrin bound iron (NTBI) is released into the plasma [15,16]. This free iron, particularly a species known as labile plasma iron (LPI), generates reactive-oxygen species resulting in organelle damage and cell death, especially of hepatocytes, cardiomyocytes and the cells of endocrine glands [17]. Vital organ function is disturbed in this way, leading to serious complications which may be lethal. This necessitates the daily consumption of iron chelating agents to prevent complications and ensure survival. The degree to which these effects of iron overload occur is related to transfusion dependency. In non-transfusion dependent forms of thalassemia (NTDT), such as β-thalassemia intermedia and α-thalassemia, there is also iron overload secondary to increased absorption from the gut. However, this develops at a much slower rate than in transfusion-dependent thalassemia (TDT) [18]. Complications in these NTDT appear later in life, mostly in the second and third decades (the clinical effects of NTDT are summarized below).

## 3. Clinical Considerations

According to the causative genetic defect, the thalassemia syndromes are usually classified as β- or α-thalassemias. Here, in Table 1, we attempt to use a classification according to clinical severity, which may include several genetic types in one category.

Alpha thalassemia hydrops fetalis is caused by deletion or inactivation of all four α globin alleles. The result is that excess gamma globin chains form tetramers (γ_4_ = Hb Bart’s) in uterine life, which because of their high oxygen affinity cannot effectively deliver oxygen to tissues. This leads to severe hypoxia [19]. Intrauterine anemia leads to heart failure although the underlying mechanisms are still to be fully understood [20]. There are signs of pronounced fetal edema, hepatosplenomegaly and hydramnios. Maternal pre-eclampsia and the need for caesarean section endanger mother’s health and life. These possible outcomes have made it a rule that prenatal diagnosis is offered to at-risk pregnancies with termination of pregnancy before maternal health is affected. However, intra-uterine blood transfusions have led some pregnancies to a successful outcome and the babies to survive as transfusion dependent patients [21]. The molecular defects that can cause hydrops include deletional mutations found in the Mediterranean countries like --^MED^/--^MED^, but more commonly in Asia like --^SEA^/--^SEA^; in addition, in South East Asia severe non-deletional mutations are more common, and so hydrops is more commonly encountered in that region. Some of these non-deletional mutations correspond to a mixed defect in which the thalassemic determinant is associated with an unstable abnormal hemoglobin. Examples are Constant Spring (common), Quong Sze, Suan Dok, Pakse, and Adana (rare) hemoglobin [22].

TDTs are the most serious clinical entities which become clinically apparent in infancy and result from β^0^ or severe β^+^ homozygosity. Triplication or quadruplication of α genes aggravate β-thalassemia and can even transform a classically asymptomatic β-thalassemia heterozygosity into a clinically relevant condition.

The hallmark of TDT is a steadily progressive anemia which makes the child transfusion-dependent from the first few months of life. The onset of clinical symptoms coincides with the fetal to adult hemoglobin switch in which HbF production decreases and is normally replaced by the production of HbA. However, because of the thalassemic defect, the switch is either abolished (β^0^/β^0^ homozygosity) or the production of HbA is grossly insufficient to compensate for the HbF decrease (β^0^/β^+^ or severe β^+^/β^+^). The continuous fall of the hemoglobin level and all the consequences described above lead to the need for repeated blood transfusions. The therapeutic aim is to keep a level of hemoglobin that will not only ensure good oxygenation, but also reduce the stimulus for EPO secretion and thus reduce endogenous erythropoiesis [23]. Keeping the pre-transfusion hemoglobin above 9–10 g/dL achieves this aim and allows for physical development with reduced or no bony changes and deformities. Regular lifelong transfusions have several possible adverse effects which include immunological reactions and transmission of infectious agents of which the hepatitis C virus is currently the most common. In many countries, inadequate supplies of donors result in low hemoglobin levels, with the consequences of anemia described before. The most important side effect, however, is the accumulation of iron, the pathophysiological consequences of which have been described above. The life endangering effects of iron toxicity necessitate close monitoring and quantification of the iron load in the tissues and removal of the iron by iron chelating agents [24]. The thalassemia patient is therefore subjected to a series of tests aiming at prevention or at least early recognition of tissue toxicity. 

Regular monitoring of regularly transfused patients includes
Regular blood tests: hematology, biochemistry and serologyImaging: MRI (to measure heart and liver iron load), abdominal ultrasound, bone densityEchocardiography to assess cardiac function and pulmonary hypertensionOphthalmological examinations and audiometryOrgan biopsies as required (largely replaced by MRI)

On a global level, effective iron chelation is hampered by:Poor availability of drugs in many countries and catastrophic out of pocket expenses [25].Patient non-adherence to prescribed treatment [26]. In many clinics, non-adherence is regarded as a major cause of treatment failure. This is not surprising since chelation treatment is a daily routine, and any short or long interruption leads to the exposure of cells to free iron radicals with consecutive tissue damage. Various interventions have been suggested to reduce this phenomenon mainly relying on psychosocial support and the patient partaking in management decisions which concern them. Understanding patient concerns is still an open subject [27,28] and effective interventions are still a problem in everyday life of thalassemia clinics across the world.Inexperience and inadequate adherence of physicians to evidence based guidelines. This is a phenomenon which is not well documented in scientific publications but is a common experience where rare conditions are concerned [29]. Due to the rarity of the condition in many localities, thalassemia suffers from all the weaknesses reported by EURORDIS and other rare disease organizations, such as delayed diagnosis and recognizing life-threatening complications too late [30].

The long survival experienced by the latest birth cohorts of patients with the most severe thalassemia syndromes, mainly treated in centers of expertise, is due to a combination of safe and effective blood transfusion, adequate iron chelation and early recognition of complications with effective interventions by a multidisciplinary team of experts. The excellent outcomes in terms of survival and quality of life were not achieved by the introduction of new additional therapies [31], but by adherence to evidence-based guidelines. New treatments are expected in the near future [32] which will probably further improve quality of life. Curative treatments, apart from haemopoietic stem cell transplantation (HSCT) which has long been available [33], are also in the pipeline, utilizing genetic therapies.

Milder, non-transfusion dependent forms (NTDT), which can survive without regular blood transfusions may require occasional transfusions during intercurrent illnesses or pregnancies. However, regular transfusions may be required in later life due to complications. NTDT syndromes are caused by mild β^++^ mutations (allowing some β globin chain production) and/or the co-inheritance of mitigating factors such as α-thalassemia or persistence of fetal hemoglobin which reduce the globin chain imbalance. Another mechanism is the co-inheritance of another hemoglobin variant, the most common being HbE which is commonly encountered in South East Asia [34]. HbE is a “thalassemic variant”, i.e., a hemoglobin variant that is produced insufficiently and thus promotes anemia.

Since erythropoietic tissue expansion is not suppressed early in life by transfusions, its effects on bone marrow expansion and extramedullary erythropoiesis continues and so damage to bone structure, leading to deformities and relatively early onset of osteopenia/osteoporosis and pressure from hematopoietic masses, characterize this syndrome. Chronic anemia and ineffective erythropoiesis persist despite a relatively steady hemoglobin level with the result of increasing iron absorption from the gut through hepcidin suppression. This causes iron toxicity mainly to the liver while the heart remains relatively free of iron-related damage [35,36,37]. Iron overload with the increased circulation of NTBI leads to hepatocyte damage which progresses to fibrosis, presumably due to longer duration of exposure [38] and to hepatocellular carcinoma, which is more frequent in NTDT. Erythropoietic expansion as evidenced by the respective markers like soluble transferrin receptors (sTfR), increased nucleated red cells (NRBCs) and growth differentiation factor 15 (GDF-15) [39], will also contribute to splenomegaly leading many clinicians to recommend splenectomy. The pathophysiological effects of splenectomy and hypercoagulability are more often encountered in NTDT compared to TDT and so stroke, pulmonary hypertension and vascular disease are more frequent in NTDT compared to regularly transfused patients. It is because of these complications that splenectomy is avoided and that regular transfusions are initiated in NTDT patients when complications arise such as symptomatic extramedullary hematopoietic masses, pulmonary hypertension, thrombotic events, and leg ulcers [40]. Interestingly, causative factors of NTDT are variable, including mostly β-thalassemia intermedia and HbE/β-thalassemia but also HbH disease (which belongs to the α-thalassemia group). There are differences in both the biomarkers of iron metabolism and erythropoiesis as well as in the clinical manifestations among the various causative factors [39]. For instance, HbH disease in the Mediterranean has the mildest effects but can be severe in South East Asia. Similarly, for unknown reasons, HbE/β-thalassemia presentation varies from relatively moderate to very severe.

## 4. Management of the Thalassemia Syndromes: The Global Perspective

It is not possible here to go over the details of all treatment modalities and their possible effectiveness or side effects. The impression in many high prevalence areas is that providing adequate supplies of ‘clean’ blood and a choice of iron chelating agents is the basis of managing these syndromes effectively. This is particularly true if the thalassemia population consists of children. From adolescence and even earlier, a monitoring schedule should be in place, aiming to recognize early complications which should be dealt with. Centers, mainly in the economically developed world, which are able to fully follow internationally accepted guidelines [41] are serving a minority of the global community of patients [1,3,4]. Such privileged patients are now surviving to their fifties with a good quality of life. Even in locations with few resources, essential components of care cannot be ignored or put aside because of “other priorities”. The reason is that any reductions will increase the chance of complications and so increase the cost of care and/or result in premature death. This is a waste of resources that is often not recognized. The burden of disease, in the case of congenital disorders, cannot be simply assessed by the numbers of patients affected. In the reports of the Global Burden of Disease (GBD), thalassemia was ranked 68th in 2010 in terms of DALYs, yet it ranked 24th when the one-to-four-year age group was considered, indicating its important contribution to under-5 mortality [42]. It is doubtful whether this ranking is based on accurate data since many children in countries with high prevalence of thalassemia and with less privileged populations may die without even a diagnosis. National policies are usually formulated by public health officials who have no clinical experience and use DALYs and GBD data to rank their country’s priorities. The hemoglobin disorders are therefore not regarded as a priority. Late-onset diseases such as cardiovascular disorders and diabetes are given high ranking and are prioritized, even by WHO in its non-communicable disease (NCD) program, leaving congenital and hereditary disorders with no plan either for patient survival or even for prevention. The results are disastrous, and early death often makes the problem invisible [4]. Any improvements in thalassemia management will benefit health services for many needs in the community:Adequacy of blood supplies—Regularly transfused patients require more blood than the general population and so blood collection drives, donor education, and good practices in donor management are organized. These efforts which aim to have adequate supplies will benefit the whole community and patients who require blood transfusion circumstantially for whatever reason will also benefitSafe blood—Regularly transfused patients are at higher risk from contaminated blood from both bacteria and viruses, and in some locations malaria is also a threat. Having strict screening procedures to screen donors will make blood safer for all the community.Reactions to blood transfusion are more common in regularly transfused patients, especially alloimmunization. Having procedures and technology for leukodepletion and extended antigen typing (including molecular typing) in place, will help many patients in the community (N.B., regularly transfused patients are not only those with hemoglobin disorders but include other congenital anemias, myelodysplasias, and bleeding disorders).Having availability of quality medication, so that effectiveness and safety of drugs is guaranteed, is a universal requirement. As generic drugs are increasingly becoming available and affordable, their quality should be more strictly controlled. This will help all patients, especially those with life-long dependency due to chronic disease.Centers of expertise are healthcare facilities where standards of care for chronic and rare diseases can be guaranteed. Coordinated multidisciplinary teams have been shown to improve patient outcomes where multi-organ disorders are concerned [43]. Centers of expertise can support other centers with fewer patients and less experience in an organized and officially recognized networking system. This is a universal recommendation supported by a system of accreditation of centers. This concept has been recognized by the European Commission through projects like EURORDIS and ENERCA which has resulted in criteria for centers of expertise [44,45] and the creation of European Reference Networks (ERNs) for rare diseases including rare anemias. The Thalassemia International Federation (TIF) is now developing disease-specific standards aiming at the accreditation of centers as a means for quality improvement.Following evidence-based guidelines is another universal recommendation.Universal health coverage will ensure that families are not bankrupted by the demands of a chronic lifelong condition. Out of pocket expenses are the major reason why in some countries optimum care is not accessible for all patients—with all the known consequences. “Health is a human right. No one should get sick and die just because they are poor, or because they cannot access the health services they need” (Dr. Tedros Adhanom Ghebreyesus, Director General WHO, World Health Day 2018 Advocacy Toolkit; 7th April 2018; https://www.who.int/campaigns/world-health-day/2018/World-Health-Day-2018-Policy-Advocacy-Toolkit-Final.pdf?ua=1; last access: 20 March 2019)

## 5. Prevention and Screening

Prevention programs have reduced the birth prevalence of thalassemia in some countries and possibly saved resources for patient care. Such programs require planning and investment in order to include public awareness, screening to identify carriers, genetic counselling aiming to assist couples in making informed choices, and finally making available solutions such as prenatal diagnosis [46]. There are considerable differences in the attitude of people towards screening as well as for prenatal diagnosis and termination of pregnancy. Cultural, religious, ethical, and legal considerations must be considered in each country, but also, in this era of increasing population mix, different attitudes within communities in any country have to be considered in planning services [47]. Even though prevention has been shown to be cost-effective [48] very few countries have adopted nationally planned programs.

Neonatal screening to identify thalassemia syndromes early is not of great benefit especially in high prevalence areas where full prevention programs are in effect, since the clinical manifestations and transfusion dependency appear early in life. Where neonatal screening for sickle cell disease is established, some thalassemia homozygote cases and hemoglobin variants can be identified using the same laboratory techniques (mainly high-pressure liquid chromatography [48] and/or capillary electrophoresis and even isoelectric focusing). However, in many countries’ patients are not identified and/or there is no patient registry on a national level and so the numbers are not known. In these situations, neonatal screening (when universal) can be useful in collecting more accurate data than surveys which often include small cohorts of a population:Hemoglobin variants can be accurately identified and so result in the well-known benefits of screening for sickle cell disease and epidemiological data on other variants can be obtained.Epidemiology of α-thalassemia can also be obtained through the detection of Hb Bart’s [49].Thalassemia major can be identified in neonatal blood by using a cut off value of 1.5% HbA [50,51].

These tests are useful for secondary prevention and epidemiological studies, especially if supplemented by molecular studies. Other forms of technology such as tandem mass spectrometry may even be sensitive enough to identify β-thalassemia heterozygotes [52].

## 6. Conclusions

The thalassemia syndromes are hereditary disorders with a complex pathophysiology and serious multi-organ involvement. Current treatment may lead to long survival and a good quality of life. This includes benefitting from a full education, marriage, and parenthood, as well as contributing to the society as ordinary citizens do. In contrast, for the majority of patients, access to quality and holistic care is not possible. For these patients, thalassemia is a tragic disease with life-threatening complications which imply death in adolescence or early adulthood and result in a life of disability. Even in well-organized and well-resourced health services the provision of adequate supplies of safe blood and iron chelation are thought to meet patient needs, often ignoring the role of endocrine, cardiac, and liver monitoring by specialized teams which can deal with emerging vital organ dysfunction. The need for at least one expert reference center supporting secondary centers within each country in an organized network must be part of a policy directed and supported at the central level. This is in accordance with the concept of European Reference Networks (ERNs) for rare disorders; thalassemia falls into this category of disease in most countries.

Emerging new therapies, such as genetic interventions aiming to reduce globin chain imbalance, are likely to benefit those able to afford current management modalities leaving the “silent majority” to struggle with what basic treatments that they can afford. The solution is for health authorities to meet their obligations to those born with these conditions and persuade society and economists that investment in their health is meeting an obligation to human rights.

In this picture the question is whether neonatal screening programs can effectively contribute to achieve the desired outcomes. In areas where effective pre-conceptual or premarital prevention programs are fully applied, few cases will be picked up postnatally. In such a setting, new-born affected infants are often in families which have been informed and have chosen to give birth to an affected child. Even where there is no prevention policy, infants generally present clinically at a very young age and require immediate intervention due to severe anemia. To develop a policy solely for the early detection of thalassemia does not seem necessary. However, where there is a program for the detection of sickle cell disease, some thalassemia syndromes and most variants may be identified. This can be beneficial for secondary prevention but also in some settings for the early detection of new cases.

## Figures and Tables

**Table 1 IJNS-05-00016-t001:** Thalassemia groupings according to clinical severity.

α-Thalassemia hydrops fetalis	Leads to death in utero in most cases
Transfusion-dependent (β) thalassemia	Leads to death in early infancy unless treated
Non transfusion-dependent thalassemia	Occasional blood transfusions required (may become transfusion-dependent in later life)
Thalassemia minor	Mostly heterozygotes for thalassemia genes (carriers), but may include some homozygotes/compound heterozygotes for very mild β-thalassemia mutations and HbE

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
