# Peer review of "Thalassemias: An Overview"

_2409-515X, 2019, doi:10.3390/ijns5010016_

Reviewer 1 Report

The weaknessesses and incompleteness of this article, along with the very poor english, make it unsuitable for publication in its present form.  Much of the information is incomplete or inaccurate, particularly the ignorance of what has been learned and published from neonatal screening programs.   Important references are left out.  syndromes are oversimplified and incomplete.   Alpha that major is not always fatal in utero; in fact some cases have been detected by NBS, transplanted, and are living.  I invite a complete resubmission

Author Response

The weaknessesses and incompleteness of this article, along with the very poor english, make it unsuitable for publication in its present form.  Much of the information is incomplete or inaccurate, particularly the ignorance of what has been learned and published from neonatal screening programs.   Important references are left out.  syndromes are oversimplified and incomplete.   Alpha that major is not always fatal in utero; in fact some cases have been detected by NBS, transplanted, and are living.  I invite a complete resubmission

 Answers:

The impression of ‘incompleteness’ is probably a result of the synoptic nature of our article. What to put emphasis on and what to leave out is not an easy choice but we went through the text very carefully and tried to give a better overview. We have also revised the English language, in particular by shortening sentences. The inaccuracies in terms of alpha thalassemia hydrops fetalis syndrome were corrected and references adapted. The impact of neonatal screening for thalassemia is negligible in the vast majority of countries in the world. For this reason, we felt that it was inappropriate to put a focus on neonatal screening and left the manuscript like it is.

Beyond that, the comments of reviewer #1 were very fundamental, unspecific and not very constructive. For that reason, we focused on the other reviews to improve the quality of our article.

 Reviewer 2 Report

This is a superb presentation of thalassaemias that fits very well with the special issue theme of newborn screening. The medical and genetic information are current and concise. The public health implications and the call for public policy action have an appropriate mix of passion and evidence.

Critique:Refocus the Abstract to emphasize the best features of this very nicely-done article.  The long paragraph on global public health prioritization (pages 5-6) has powerful and relatively fresh ideas that deserve mention in the Abstract: "[neglecting] essential components of care  ... will lead to a waste of resources"  "GBD ... is ranked 24th when the 1-4year age group is considered ... " and this is probably an underestimate of the impact of thalassaemia because of underreporting.  "early death often makes the problem invisible".  The excellent concise review of medical and genetic aspects of thalassaemia are necessary but not fresh material, and probably need less space in the Abstract.

Prevention and Screening, page 6  Recommend including at least one mention that there are different ethical, cultural, and religious approaches to prevention and screening.

Minor editorial critique, typographic, or writing style.:

Introduction, 3rd paragraph:  Political and public health terminology probably prefers "low resource" instead of "poor resource"

Intro, page 2 Please clarify the paragraph on hepcidin. The first sentence:seems to be a fragment, and looks more like a subheading.  Insert comma after "ferriportin". Please clarify:  "In severe anaemia, the cause is not recognized and increased iron absorption is the result."  Does this mean "not recognized" by the body? mechanism not known to science?

Intro, page 2,  This sentence could be shortened or divided into 2 sentences: "The degree of anaemia and severity of the clinical effect can be modified by other mitigating factors, the most common of which is the co-inheritance of factors that reduce globin chain imbalance such as when α thalassaemia is co-inherited in β thalassaemia homozygotes, resulting in a milder β thalassaemia syndrome."

Genetic Control, page 3.  "These belong to a large family of proteins, the transforming growth factor β (TGF-β) [24]."  Probably add the word "family" at the end.

 page 4 "On a global level effective iron chelation is hampered by" Perhaps this section should be formatted as bullet points

page 5 "It was not by the introduction of new additional therapies [37] even though new treatments are expected in the near future [38]. Curative treatments, apart from HSCT which has long been available [39], are also in the pipeline utilising genetic therapies." These sentences both contain contrasting ideas. Perhaps they could be rearranged to improve make the flow of thoughts in this paragraph: (a) long survival is due to careful multidisciplinary management of transfusions and iron chelation and supportive care for complications (b) HSCT is curative and is not new (c) genetic therapy and other new approaches are in the pipeline.

page 5  "The pathophysiological effects of splenectomy along with the effects of the hypercoagulable state are more often encountered in NTDT and so stroke, pulmonary hypertension and vascular disease are more frequently encountered compared to regularly transfused patients."   Is the comparison for patients with  splenectomy vs TDT without splenectomy, or NTDT vs TDT.

page 6 "Centres can support centres with fewer patients and less experience in an organised and officially recognised networking system. This is a universal recommendation supported by a system of accreditation of centres"   Perhaps provide a reference that describes these two types of centres and their accreditation.

page 7 reference #3  Weatherall is misspelled.

page 8 reference #37   Blood is written twice.

page 9 Reference #45  Mussalam km should be KM.

Author Response

This is a superb presentation of thalassaemias that fits very well with the special issue theme of newborn screening. The medical and genetic information are current and concise. The public health implications and the call for public policy action have an appropriate mix of passion and evidence.

Answer:
This review was the most helpful, pointing out the issues that should reconsidered. All points have been noted and adopted in the review (please see below).

Critique:Refocus the Abstract to emphasize the best features of this very nicely-done article.  The long paragraph on global public health prioritization (pages 5-6) has powerful and relatively fresh ideas that deserve mention in the Abstract: "[neglecting] essential components of care  ... will lead to a waste of resources"  "GBD ... is ranked 24th when the 1-4year age group is considered ... " and this is probably an underestimate of the impact of thalassaemia because of underreporting.  "early death often makes the problem invisible".  The excellent concise review of medical and genetic aspects of thalassaemia are necessary but not fresh material, and probably need less space in the Abstract.

The abstract has been adapted according to reviewer #2’s suggestions.

Prevention and Screening, page 6  Recommend including at least one mention that there are different ethical, cultural, and religious approaches to prevention and screening.

Done.

Minor editorial critique, typographic, or writing style.:

Introduction, 3rd paragraph:  Political and public health terminology probably prefers "low resource" instead of "poor resource"

Done.

Intro, page 2 Please clarify the paragraph on hepcidin. The first sentence seems to be a fragment, and looks more like a subheading.  Insert comma after "ferriportin". Please clarify:  "In severe anaemia, the cause is not recognized and increased iron absorption is the result."  Does this mean "not recognized" by the body? mechanism not known to science?

We revised the paragraph accordingly and hope that it is unambiguous now.

Intro, page 2,  This sentence could be shortened or divided into 2 sentences: "The degree of anaemia and severity of the clinical effect can be modified by other mitigating factors, the most common of which is the co-inheritance of factors that reduce globin chain imbalance such as when α thalassaemia is co-inherited in β thalassaemia homozygotes, resulting in a milder β thalassaemia syndrome."

Done.

Genetic Control, page 3.  "These belong to a large family of proteins, the transforming growth factor β (TGF-β) [24]."  Probably add the word "family" at the end.

We have deleted the whole paragraph on genetic control not required to understand the basics of thalassemia.

page 4 "On a global level effective iron chelation is hampered by" Perhaps this section should be formatted as bullet points

Done.

page 5 "It was not by the introduction of new additional therapies [37] even though new treatments are expected in the near future [38]. Curative treatments, apart from HSCT which has long been available [39], are also in the pipeline utilising genetic therapies." These sentences both contain contrasting ideas. Perhaps they could be rearranged to improve make the flow of thoughts in this paragraph: (a) long survival is due to careful multidisciplinary management of transfusions and iron chelation and supportive care for complications (b) HSCT is curative and is not new (c) genetic therapy and other new approaches are in the pipeline.

Done.

page 5  "The pathophysiological effects of splenectomy along with the effects of the hypercoagulable state are more often encountered in NTDT and so stroke, pulmonary hypertension and vascular disease are more frequently encountered compared to regularly transfused patients."   Is the comparison for patients with  splenectomy vs TDT without splenectomy, or NTDT vs TDT.

This is a very important comment. We revised the paragraph accordingly! We believe that it is unambiguous now.

page 6 "Centres can support centres with fewer patients and less experience in an organised and officially recognised networking system. This is a universal recommendation supported by a system of accreditation of centres"   Perhaps provide a reference that describes these two types of centres and their accreditation.

Done.

page 7 reference #3  Weatherall is misspelled.

page 8 reference #37   Blood is written twice.

page 9 Reference #45  Mussalam km should be KM.

All references were adapted.

Many thanks to reviewer #2 for the important comments. We accepted all suggestions and believe that they were very important for the quality of the manuscript.

 Reviewer 3 Report

The authors present an interesting and insightful review of the thalassaemia syndromes and their potential impact on health and well-being particularly in low income countries.

The information provided is not new and therefore does not contribute to scientific understanding in the field although there are some views from the authors in relation to setting health priorities in low income countries.

In general the article is well written although there is a frequent use of abbreviations without explanation throughout the text and those from a non-haematological background might find this confusing or eg NTDT, EPO mRNA and STATs are used without explanation.

There is little commentary about the role of screening in this context and given the audience of the IJNS this could perhaps be addressed.

Space permitting I would recommend that the paper is accepted although it does not offer a great deal of scientific merit.

Author Response

The authors present an interesting and insightful review of the thalassaemia syndromes and their potential impact on health and well-being particularly in low income countries.

The information provided is not new and therefore does not contribute to scientific understanding in the field although there are some views from the authors in relation to setting health priorities in low income countries.

Many thanks for these nice word. As we have already commented, it is in the nature of this review that it is just meant to give an introduction to thalassemia for the non-hematologist readership of this special issue of the IJNS.

In general the article is well written although there is a frequent use of abbreviations without explanation throughout the text and those from a non-haematological background might find this confusing or eg NTDT, EPO mRNA and STATs are used without explanation.

Thanks for this very important comment. We have explained all abbreviations in the revision.

There is little commentary about the role of screening in this context and given the audience of the IJNS this could perhaps be addressed.

Thanks for this comment and it is right that there is few information on the role of thalassemia screening. However, we did this on purpose, since thalassemia is no classical newborn screening disease, because it does not fulfil the Wilson-Jungner criteria. In particular, the benefit of early detection of thalassemia is low. Accordingly, newborn screening is of minor importance in the history of diagnosis and treatment of thalassemia. We have adapted the text at some points to clarify that issue.

Space permitting I would recommend that the paper is accepted although it does not offer a great deal of scientific merit.

Thank you very much! We highly appreciate that.